# Optical skyrmions from metafibers with subwavelength features

Tiantian He[1,2,6], Yuan Meng [1,5,6], Lele Wang[1,2], Hongkun Zhong[1,2], Nilo Mata-Cervera [3], Dan Li[1,2], Ping Yan [1,2], Qiang Liu[1,2], Yijie Shen [3,4] ✉ & Qirong Xiao [1,2] ✉

Optical skyrmions are an emerging class of structured light with sophisticated particle-like topologies with great potential for revolutionizing modern informatics. However, the current generation of optical skyrmions involves complex or bulky systems, hindering the development of practical applications. Here, exploiting the emergent "lab-on-fiber" technology, we demonstrate the design of a metafiber-integrated photonic skyrmion generator. We not only successfully generate high-quality optical skyrmions from metafibers, but also verify their remarkable properties, such as topology switchability and topology stability with subwavelength polarization features beyond the diffraction limits. Our flexible fiber-integrated optical skyrmions platform paves the avenue for future applications of topologically-enhanced remote super-resolution microscopy and robust information transfer.

Skyrmions, as a kind of topological quasiparticles with sophisticated spin textures, originated from particle and solid-state physics as emerged information carriers in local data storage[1–4]. Following their magnetic counterparts, skyrmions have recently been constructed in photonics realm through structured electromagnetic fields, namely optical skyrmions[5]. They have emerged as promising candidates to revolutionize long-range topological information transfers and light-matter interactions thanks to their abundant intriguing physical attributes[5–7]. However, current optical skyrmions generators necessitate either complex or bulk systems hindering further applications, given the challenges to generate intricate structured light fields with sophisticated polarization distributions via conventional methods. The initial optical skyrmions were constructed by harnessing evanescent electromagnetic fields via surface plasmons[8–10]. Soon after, optical skyrmions with tailored topological properties have also been generated using free-space optical vector fields in complex spatial light modulation and interferometer systems[11–16]. The free-space skyrmionic beams with spin-orbit coupling and tunable topological textures[17–19]

showcase unique features such as topological stability[20], topological state transition in disorder media[21], structural stability in nonlinear conversion[22] or resilience in free-space propagation[23,24], exhibiting great potential as robust information carriers for next-generation optical information networks. However, current experimental demonstrations of such optical skyrmionic beams require the use of bulky devices such as digital micromirrors and spatial light modulators[12,16], or cascaded integral optical elements[25]. A compact, flexible and integrated micro-generator for diverse optical skyrmions, which is thus highly desired to unlock practical applications, is still elusive to the best of our knowledge.

As a cornerstone underpinning modern communication systems, optical fibers[26] have been established as the prime choice for long-distance and high-capacity data transmission[27] due to their distinctive features, such as high aspect ratio, great flexibility, broad bandwidth, and immunity to electromagnetic interference. Compared to current systems that generate subwavelength structured light[28–30], which typically have large optics, fiber optics exhibit tremendous potential to

[1]Department of Precision Instrument, Tsinghua University, No.1 Qinghua Garden, Chengfu Road, Haidian District, Beijing 100084, P.R. China. [2]State Key Laboratory of Precision Space-time Information Sensing Technology, No.1 Qinghua Garden, Chengfu Road, Haidian District, Beijing 100084, P.R. China. [3]Centre for Disruptive Photonic Technologies, School of Physical and Mathematical Sciences & The Photonics Institute, Nanyang Technological University, Singapore 637371, Singapore. [4]School of Electrical and Electronic Engineering, Nanyang Technological University, Singapore 639798, Singapore. [5]Present address: Mechanical Engineering and Materials Science, Washington University in St Louis, St Louis, MO 63130, USA. [6]These authors contributed equally: Tiantian He, Yuan Meng. ✉e-mail: yijie.shen@ntu.edu.sg; xiaoqirong@mail.tsinghua.edu.cn

realize flexible optical device integration with decent miniaturization, flexibility, and lightweight. Driven by the maturation of "lab-on-fiber" technology[31], the integration of multifunctional nanostructures to fiber facets has become a hotspot with high degrees of freedom in design, integration handiness, and versatile functionalities. Current approaches typically leverage 3D printing[32–34] to fabricate the same-scaled-down structured devices on the fiber tip directly. However, this strategy suffers the shortcomings of limited processing precision, restricted numerical aperture (NA)[33] and single device functionality[32], rendering them incapable of realizing the desired skyrmion excitation which demands fine subwavelength structure resolution to enable intricated light field modulations. In contrast, metasurfaces[35–37] consisting of subwavelength nanoscatterers have proven excellent paradigms to offer precise, powerful, and multi-dimensional control over the amplitude, phase, and polarization of light, establishing itself as a remarkable choice to facilitate the miniaturization and creation of high-dimensional structured light. So far, metadevices have shown generators of conventional structured light[38–40], but have not yet been applied for topological optical skyrmions.

Here, we report the on-demand generation of various optical skyrmions via fibers-integrated metadevices to the best of our knowledge. By judiciously designing the polarization-dependent metasurface resting on a fiber-tip, two basis beams (one containing angular momentum) are assigned on two orthogonal polarizations. The space-polarization inseparability of the beams occurs in the near-field of fiber facet, inventively exciting the optical skyrmion. In this paper, a zeroth-order Bessel beam (BB) and a first-order BB with high NA up to 0.8 in linear polarized states are chosen as demonstrations. It is worth mentioning that our proposed device can be dynamically switched between "on" and "off", i.e. it can be tuned from non-skyrmion to skyrmion states generation by adjusting the polarization angle of input light, providing sufficient flexibility for practical applications. We also verified the topological transformation properties of the generated skyrmions, confirming remarkable topological

properties from the excited skyrmions. In addition, we experimentally demonstrate the effective high-quality excitation and modulation of skyrmions in subwavelength scale, achieving high skyrmion number up to 0.97, revealing a nearly-full skyrmion map on the Poincaré sphere. Moreover, the subwavelength polarization features are also experimentally verified to be down to ~$\lambda/5$ ($\lambda$ represents light wavelength), paving the way for future ultrahigh-density and ultrastable information storage and transfer. Our metafiber skyrmion scheme significantly increases flexibility and integration, holding great promise for exploring topological photonics, optical communications, and super-resolution microscopy.

## Results
### Concept

Optical skyrmion is a quasiparticle state of light with distinctive topological vector textures, and it can be constructed by polarization Stokes vectors of a customized vector beams. A typical class of such skyrmionic beams[11,12,17] can be defined as,

$$|\psi\rangle = |\ell_1\rangle|e_1\rangle + e^{-i\Delta\varphi}|\ell_2\rangle|e_2\rangle \tag{1}$$

where $\ell_1$ and $\ell_2$ represent the orbital angular momentum (OAM) charge of the scalar vortex beams with two orthogonally polarization states $|e_1\rangle$ and $|e_2\rangle$ respectively; they can be either Lagrange-Gaussian (LG) modes or Bessel modes (here Bessel modes are chosen in this paper). The construction and properties of optical skyrmions based on the circular polarizations are schematically shown in the inset box of Fig. 1. By superimposing two modulated orthogonally polarized Bessel beams (BBs): zeroth-order BB ($J_0$) and first-order BB ($J_1$) carrying OAM[7], in right-handed and left-handed circular polarizations (RCP and LCP) respectively, a Stokes skyrmion can be constructed (Fig. 1a). The resultant spatially-varying vector pattern will fulfill a skyrmion mapping (Fig. 1b), i.e. the mapping of all polarization states from a parametric sphere to a localized transversal plane[17]. To visualize this

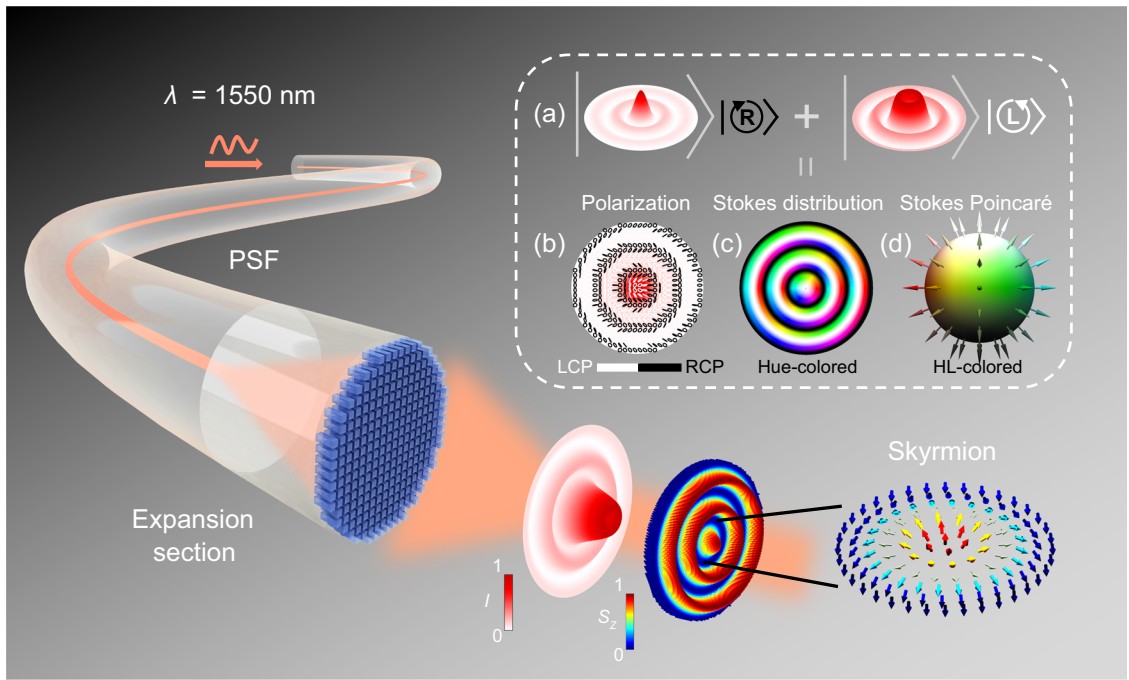

**Fig. 1 | The schematic diagram of the skyrmions' excitation from metafiber.** The 3D intensity (*I*) and vector distribution of the excitation skyrmion are shown after exiting the metafiber. $S_z$: z-component of Stokes vector, PSF: polarization-maintaining single-mode fiber, $\lambda$: light wavelength. Inset: **a** The construction principle of skyrmion, composed by zeroth-order Bessel Beam and first-order Bessel Beam under orthogonal left circular polarization (LCP) and right circular polarization (RCP). **b–d** The properties of the excited skyrmion. **b** The elliptical polarization distribution with underneath electric field intensity. **c** Hue-colored Stokes polarization distribution. **d** Hue-lightness (HL)-colored unit Stokes Poincaré sphere.

mapping, we utilize hue-lightness (HL) colors to label states of polarization: Each polarization state corresponds to a point on the Poincaré sphere and is represented by a unit normal Stokes vector ($S_x$, $S_y$, $S_z$) at the surface of the sphere, called Stokes vector, which embodies physical properties for polarized light. Lightness color (from black to white) is applied to represent the longitudinal component $S_z$ (from −1 to 1 or down to up; represent the preponderance of RCP light over LCP light) and use hue color to visualize the azimuth of transverse ($S_x$, $S_y$) components. Thus, the skyrmions can be identified by the HL-colored map of corresponding vector fields. For instance, in the HL map of the Bessel-based skyrmionic vector beam (Fig. 1c), there is a full HL color area at the center region, which shows a stereographic mapping to the Poincaré sphere (Fig. 1d). A perfect Stokes skyrmion will cover the full polarization states, meaning a full coverage of Stokes Poincaré sphere.

The schematic diagram of the metafiber-based skyrmion generators is illustrated in Fig. 1, consisting of a polarization-maintaining single-mode fiber (PSF), an expansion section, and a single-layer metasurface on the fiber-tip. The metasurface is designed to independently modulate the phase of the two orthogonal polarized lights when passing through the meta-tips. Hence, the BBs are tightly focused and inseparably entangled to generate a Stokes skyrmion in subwavelength scale, enabling a robust propagation within a non-diffracting distance in free space. The intensity profile and Stokes vector distribution of the whole light profile are shown in right-bottom panels of Fig. 1, and a corresponding hedgehog-like Stokes vector configuration is illustrated (enlarged), which points up and down in the center and the edge of the confinement region of the localized skyrmion.

## Simulation and manipulation of skyrmions

To construct optical skyrmions, two different scalar beams should be judiciously modulated on orthogonally polarized basis, one of which carries OAM. The polarization basis can be chosen as, for instance, the horizontal ($x$) and vertical ($y$) linear polarizations, as well as the RCP and LCP bases. The change on the polarization basis will only change the polarization texture, but not affect the topological properties and the skyrmion number. In this study, two orthogonal linearly polarized states are chosen as basis. We demonstrated to modulate the $J_0$ in the $x$-pol and the $J_1$ in the $y$-pol to excite a Stokes skyrmion. The $J_0$ can be approximately generated by printing an axicon phase profile on a plane wave, and introducing a vortex phase into $J_0$ yields the vortex BB $J_1$. A metalens phase profile is added on the metasurface for compensating the spherical phase distribution of the light source after the expansion section. The metasurface is made of silicon nanoantennas on top of quartz substrate, with a lattice period of 700 nm for each meta-unit. The lateral size of the Si antennas ranges from 250 nm to 550 nm to ensure a full $2\pi$ phase coverage. The propagation phase concept is used for the design of dual-BBs in orthogonal polarizations. By a specific design of metasurface, dual-order BBs under linear polarizations are achieved (The design strategy of metasurface is detailed in Supplementary Note 1). Detailed description of the expanding section is also elaborated in Supplementary Note 2.

Theoretically, the transverse intensity profiles of the BBs remain constant as the propagation distance $z$ increases showing non-diffracting feature[41]. However, ideal BBs require an infinite aperture as well as infinite power, making it not practically possible to excite ideal BBs using infinite sized optical devices. Real devices with finite aperture limit the non-diffracting behavior of quasi-BBs to a certain finite distance as proven in previous works[41]. For the topological quasi-particles, skyrmion with smaller size will lead to higher information storage density. Therefore, BBs[41] with NA up to 0.8 are chosen here to exploit the potential of sub-wavelength skyrmion generation in metafibers. A longitudinal intensity profiles of $J_0$ and $J_1$ are illustrated in the first row of Fig. 2a. The intensity and phase profiles at the

transverse plane highlighted with the dashed white lines are shown in the bottom two rows of Fig. 2a. The left two columns correspond to the ideal profiles and the third and fourth correspond to the simulated ones, showing an excellent consistency. We use rigorous full-vector simulation with finite-difference time-domain method and the longitudinal electric field components are rigorously considered. When the light beam $J_0$ is superposed with $J_1$ in orthogonal linear polarizations, a Stokes skyrmion beam with bimeron texture is achieved.

The continuous modulation is a crucial property of the device, which not only facilitates a deeper comprehension of topological transformations, but also substantially elevates the degree of integration in specific applications like multilevel information encoding. The modulation of our proposed metafiber is shown in Fig. 2b–d. On the one hand, by tuning the polarization angle of the source, switching between non-skyrmion and skyrmion can be achieved, similar to the changing between non-OAM and OAM states. To clearly demonstrate this continuous transformation, a quantitative curve is presented in Fig. 2c. The absolute value of the angle between source polarization direction and $x$-axis is defined as $\theta$ in degrees. The power is calculated by integrating the intensity over the entire transverse cross-section. The total power of the source $P_t$ remains unchanged in this transformation. The pink and green lines represent the power in $x$ and $y$ polarization respectively, with the intensity profiles embedded in Fig. 2c for different input polarization angles 0°, 30°, 45°, 60°, and 90°. It is evident that the power of the $J_0$ progressively diminishes as the polarization angle increases, whereas $J_1$ steadily increases, due to the near-unity transmission of the metasurface for any polarization input. The trend can be quantitatively fitted as $P_{x,\,input} = P_t \times \cos^2(\theta)$ and $P_{y,\,input} = P_t \times \sin^2(\theta)$. Thus the polarization distributions of the superimposed beams - skyrmions change with $\theta$. Stokes vector distributions of typical output skyrmion states after a quarter wave plate (QWP) are shown in Fig. 2b. The vector shows a skyrmion distribution with a vortex texture around the center when $\theta = 45°$, while the vectors exhibit uniform polarization states in $\theta = 0°$ or 90°. On the other hand, the skyrmionic vector textures can transform smoothly without changing the topological features. In order to verify this property, we introduce a QWP to change different quasiparticles' textures. Typical vector distributions are shown in Fig. 2d. Specifically, a Bloch-bimeron texture is achieved by linear polarizations modulation in our design, and the bloch-skyrmion texture can be obtained by introducing a QWP with the fast axis at 45° with respect to the $x$-axis (corresponding to transform linear polarization basis into circular polarization basis). When the fast axis of the QWP aligns with the $x$-axis, the vector distribution of the bimeron type remains and exhibits a texture composed of two half-merons with opposite polarities. By rotating the QWP from 0° to 45°, the intermediate states appear.

The Poincaré Sphere provides a graphical visualization of the Stokes vector, allowing a clearer representation of the modulated vector transformations. The coordinate axes of the Poincaré sphere are normalized ($S_x$, $S_y$, $S_z$) (detailed in Supplementary Note 3). The skyrmion number ($N_{sk}$) here characterizes the similarity to the perfect skyrmion[8,30,42], as well as the quality of resemblance of the generated beam with ideal skyrmions (detailed in Methods). For the polarization angle regulation, the skyrmion maps on the Poincaré sphere are plotted in Fig. 2e, with the polarization angle $\theta$ and the source electric field polarization directions also labeled. When the polarization angle is 0°, the points on the Poincaré sphere are very sparse, and the skyrmion number is only 0.06, showing a non-skyrmion state. As the polarization angle increases, the vector distributions gradually spread until they fully cover the Poincaré sphere. When the polarization angle is equal to 45°, the Poincaré sphere is covered all over the sphere and contains a full HL color, demonstrating a perfect skyrmion with a skyrmion number of 0.99. In addition, our proposed device can generate a high quality skyrmion for input polarization angles between 20°–80°, which means that the topological fields can be generated

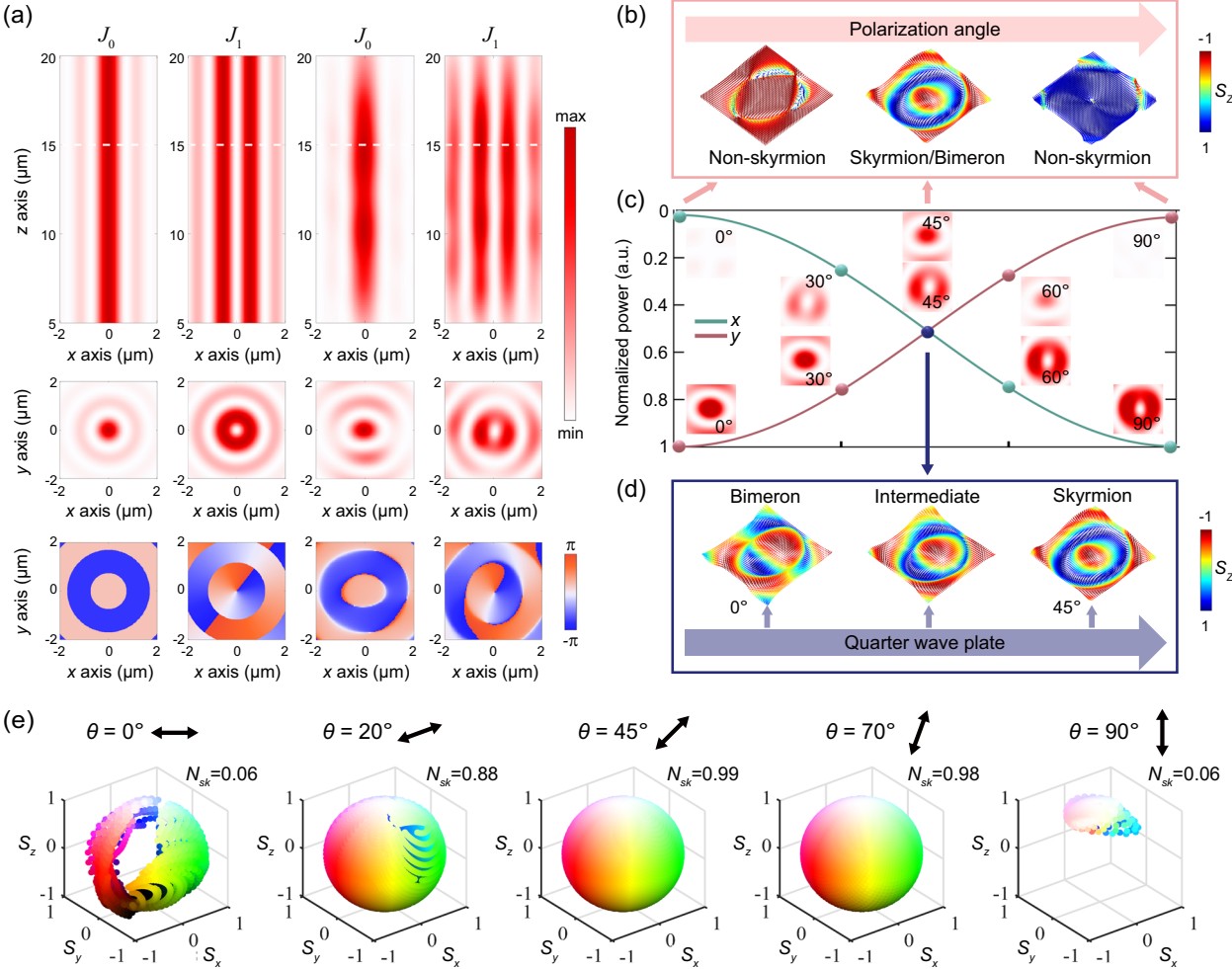

**Fig. 2 | Metafiber design for generating tunable optical skyrmions. a** Ideal and simulated light field distributions of the skyrmion with components $J_0$ and $J_1$ in orthogonal linear polarizations. The first and second columns are ideal profiles and the third and fourth are simulated profiles. The intensity profiles in the longitudinal plane $xz(y=0)$, and the intensity and phase profiles in the transverse $xy$ plane (with $z$ marked by white lines in the first-row figures) are illustrated from top to bottom. **b** The manipulation of skyrmion/non-skyrmion states via input polarization angle $\theta$.

The vector distributions of major states are shown. **c** Normalized power curves with respect to the polarization angle. The intensity profiles of $J_0$ and $J_1$ of different polarization angles are inserted and marked in lines with dots. a.u.: arb. units. **d** The manipulation of topological transformation between different quasiparticles via quarter wave plate. **e** The maps on the Poincaré sphere in the transformation between skyrmion and non-skyrmion states in different polarization angles $\theta$ in simulation, with the corresponding skyrmion numbers $N_{sk}$.

without the need of very precise adjustments, making it valuable for practical applications. Meanwhile, the texture transformations can also be characterized by means of the Poincaré sphere. The skyrmion densities and corresponding skyrmion numbers along with $\theta$ are also plotted in Supplementary Note 3. The interconversion of skyrmion and bimeron textures can be realized by QWP, which also corresponds to a coordinates transformation on the Poincaré sphere. To take a special case, when the angle between the fast-axis of a QWP and the $x$-axis is 45°, the polarization distributions are transformed and different, but the topology as well as the skyrmion number are not affected, since this transform corresponds to the change of coordinate system of the Poincaré sphere [from $(S_x, S_y, S_z)$ to $(S_z, S_x, S_y)$; detailed in Methods]. The regulation of the skyrmion texture as well as the bimeron texture with detailed $\theta$ are discussed in Supplementary Note 4. Besides, our proposed design also has a favorable broadband characteristic beyond 200 nm and a good non-diffraction property (discussed in Supplementary Note 5).

Furthermore, skyrmions with various textures are also extended to generate using our proposed scheme. With horizontal/vertical (H/V) polarization basis, the resulting beams are composed by two merons, referred to as bimeron states. Introducing a QWP converts H/V into R/L

(right/left circular), and the bimeron states are transformed into the skyrmion states. The intensity, polarization and vector distributions of some typical textures are illustrated in Fig. 3, with the bimerons and skyrmions displayed in the upper and lower rows of Fig. 3 respectively. According to Eq. (1), for the case of $\ell_1 = 0$, $\ell_2 = 1$, when $\Delta\varphi = 0$ and $\pi$, the vector field presents a hedgehog texture (vectors oriented in the radial direction), named Néel-I and Néel-II textures respectively. When $\theta = \pi/2$, a vortex texture (vectors oriented in the azimuthal direction) is obtained, named Bloch texture, which is chosen for experimental demonstration. When the $\Delta\varphi$ varies in the ranges $[0, \pi/2]$ and $[\pi/2, \pi]$, intermediate textures appear. On the other hand, by setting $\ell_1 = 0$, $\ell_2 = -1$, a saddle texture (namely anti-type) is realized. Moreover, high order skyrmions and other textures such as skyrmioniums and bimeroniums can also be generated by increasing the topological charge $\ell$ of the BB.

## Experimental results
The metasurface is manufactured by standard electron beam litho-graphy (EBL) and lift-off process (Supplementary Note 6). The meta-structures are then carefully attached to the fiber facet[26,43] to enable good beam irradiation on the metasurface, ensuring sufficient light

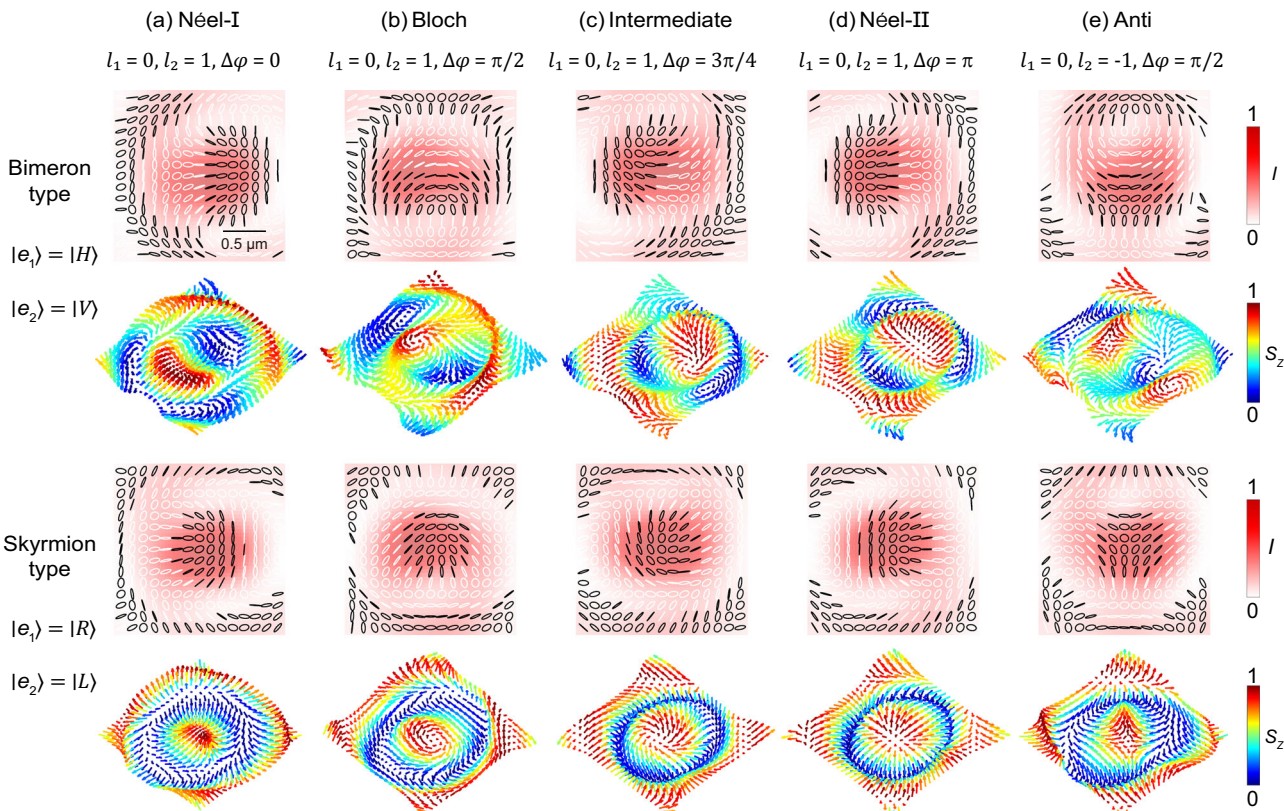

**Fig. 3 | Topological texture tunability.** Simulated elliptical polarization distributions and 3D vector distributions for various topological textures generated from different metafibers. Corresponding intensity profiles are shown under the polarization distributions. From left to right columns: (**a**) Néel-I, (**b**) Bloch, (**c**) Intermediate, (**d**) Néel-II, and (**e**) Anti textures. Upper two rows: bimeron-type, lower two rows: skyrmion-type. Scale bar: 0.5 μm.

wave modulation (see Methods). Analysis of errors in fabrication procedures and experimental processes are elaborated in Supplementary Note 7. A physical picture of the integrated device and an oblique scanning electron microscope (SEM) of the local metasurface are depicted in Fig. 4a, with a detailed chart inserted. To validate the properties of the excited skyrmion, the complex amplitudes of the light field in $x$ and $y$ polarization are required. Therefore, the experimental setup is built (detailed in Methods and Supplementary Note 8), including an amplified light path and a coherent parallel light path. Then, the intensities of the interference light fields and the plane fields are recorded. Consequently, through the phase recovery method (Supplementary Note 9), the complex amplitudes of the object fields under two orthogonal linear polarizations are obtained, as shown in the bottom of Fig. 4b, while the upper row represents simulation results. The experimental $x$-polarized intensity distribution shows a dark-red central spot surrounded by a white ring corresponding to the first zero, and the phase is nearly uniform in the beam's central region. The $y$-polarized intensity pattern has a typical doughnut shape with a white dot in the center surrounded by a dark-red ring, while the resulting helical phasefront presents a point singularity in the central region, showing a good agreement between the experimental results and the simulations.

By mapping $J_0$ in the polarization state $|H\rangle$ and $J_1$ in $|V\rangle$, the bimeron state is realized. Afterwards, introducing the half wave plate to convert the polarization basis of $J_0$ as $|R\rangle$ and $J_1$ as $|L\rangle$, the skyrmion states can be realized as well. The stokes polarization distributions of the skyrmion and bimeron states are shown on the left and right sides of Fig. 4c, respectively. From the lower left corner, skyrmion's stokes polarization distribution shows a white central region, indicating left-handed circular polarization, surrounded by a black circle representing right-handed circular polarization. A colored ring appearing

between the black and white regions reveals the coverage of all the intermediate polarization states in the Poincaré sphere, and the colors change counterclockwise according to hue color spectrum. On the other hand, the stokes polarization distribution of the bimeron state in the right corner exhibits white and black regions on the left and right sides, with varying colors in the middle, showing two merons with opposite polarity. Additionally, the skyrmion density and polarization distribution of the realized skyrmion texture are shown in Fig. 4d. The corresponding Stokes parameters distributions are shown in Supplementary Note 10. After the calculation of the topological properties as detailed in Methods, the skyrmion number in the experiment is 0.97, which is absolutely close to the value in the simulations (0.99), proving the generation of a high quality skyrmion from the metafiber. The polarization distribution of bimeron texture is also attached to Supplementary Note 11. The Poincaré Spheres of skyrmion and bimeron in experiments are illustrated in Supplementary Note 12.

Furthermore, subwavelength feature of the polarization Stokes parameters is also realized in our proposed meta-skyrmion generators. The direction of the electromagnetic, which defines the polarization and local photonic spin state of light, is not directly subject to conventional light intensity spot, and can vary at much smaller scales[9]. The variation of longitudinal stokes parameter $S_z$ with spatial position illustrates how the polarization state of light changes throughout the space. The stokes vector distributions of the skyrmion and bimeron states of experiments are inserted in Fig. 4e, and the corresponding $S_z$ curves along the red arrows are also illustrated. The simulated and experimental results are orange and ink-green, with the original data presented with increased transparency and its absolute value plotted without transparency. The polarization inversion width is defined as the distance between two points where the normalized Stokes parameter $S_z$ changes sign and reaches half its maximum value[9], indicating

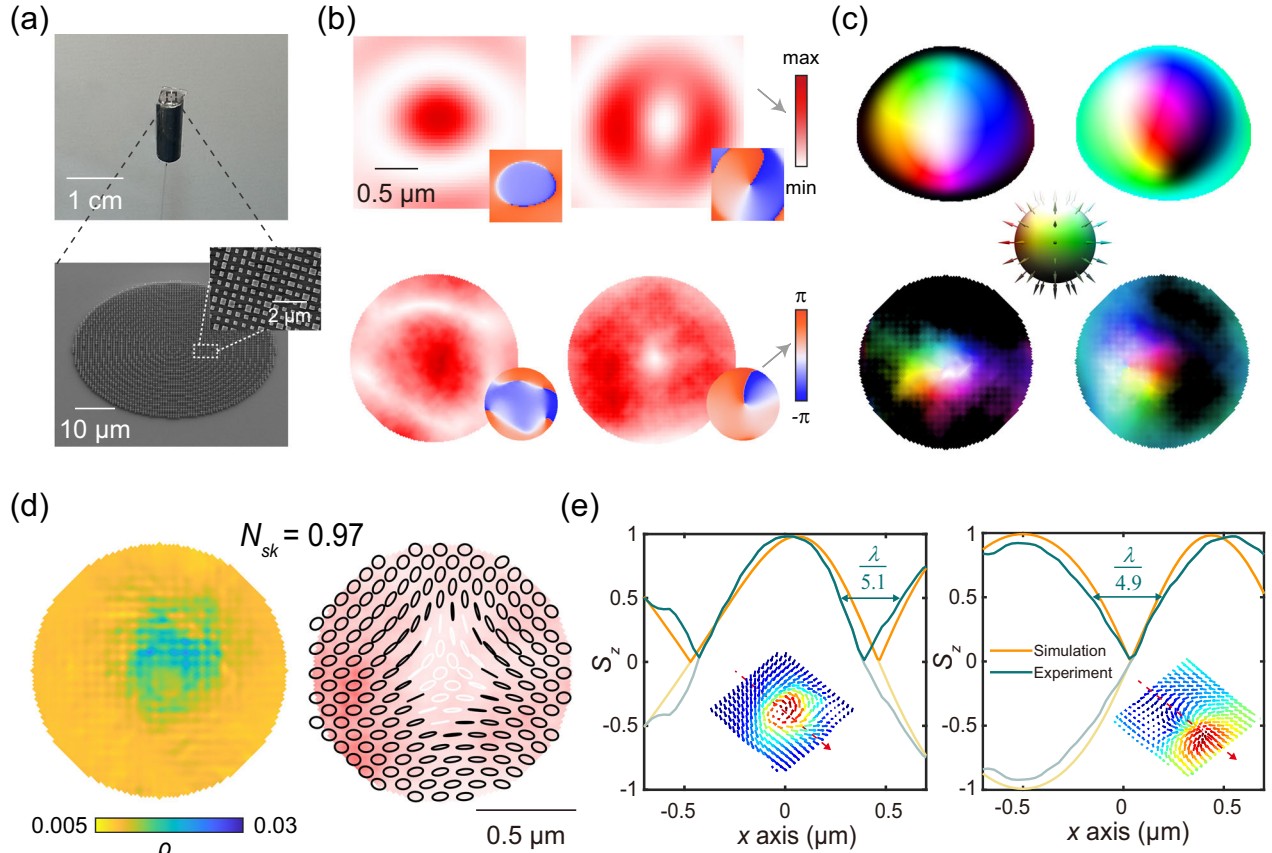

**Fig. 4 | Experimental results. a** The physical picture of the whole device (upper). The SEM of the whole metasurface (lower) and the top view of local metasurface (inset). **b** The complex amplitude profiles of the simulation (upper) and experimental results (lower). Scale bar: 0.5 μm. **c** The Stokes polarization distributions of skyrmion (left) and bimeron (right) of the simulation (upper) and experiment (lower), respectively. The colormap is in the middle. **d** The skyrmion density and polarization distribution of the skyrmion state in experiments. Scale bar: 0.5 μm. $N_{sk}$: skyrmion numbers. $\rho_{sk}$: skyrmion density. **e** The normalized Stokes parameter and its absolute value versus $x$ of skyrmion (left) and bimeron (right) states, inserted with the vector distributions of experiment results. $\lambda$: light wavelength.

the scale of the coverage of half the Poincaré sphere (from the upper-half parallel to the lower-half parallel). It can be seen that both skyrmion and bimeron reverse the polarization states with fine subwavelength features of $\lambda/5.1$ and $\lambda/4.6$ respectively, which are smaller to the diffraction limit. Therefore, in agreement with our theoretical proposal, we experimentally verify the subwavelength polarization feature of the generated Stokes topological fields, which can be of great importance for applications where miniaturization and integration is required. Good agreement is observed between simulated and experimental skyrmion textures, with very small discrepancies due to errors in device fabrication and the measurement of the electric field (discussed in Supplementary Note 13).

## Discussion

In conclusion, we have demonstrated the very compact and flexible generators of optical skyrmions using metafibers. An efficient and simple scheme is proposed to excite skyrmion in free space by judiciously designing the metafiber. The metasurface is endowed by two beams in orthogonally polarized states independently, one of which carries OAM. Either Bessel mode or LG mode can be chosen as the basis. We choose the zeroth-order BB and the first-order BB as beam basis for demonstration. By tuning the polarization angle of source, a flexible continuum modulation between the skyrmion and non-skyrmion states is realized. The transformation of textures between the skyrmion and bimeron quasiparticle states is also verified by introducing a QWP placed after the metafiber. We also experimentally demonstrate the metafiber and validate the quality of the excited skyrmions, exhibiting a high skyrmion number up to 0.97. By using a polarizer and a QWP in free space, the non-skyrmion, skyrmion and bimeron states are all realized. It is worth mentioning that the subwavelength polarization feature of Stokes skyrmions is initially proposed and experimentally verified. Compared to previous methods of spins[9] and magnetic fields[14], our proposed scheme can be detected easier and has ability to propagate to far field with excellent diffraction resilience, which can be extended more broadly to kinds of applications, such as optical tweezers, optical communication, etc. In addition, our device demonstrates remarkable stability to generate high quality skyrmions in a broad range of input polarization angles. In general OAM generations, the light field with tight focus suffers from deformation[41], disturbing the quality of the generated OAM beams. In our proposed devices, in contrast, the Stokes skyrmion with tight focus shows an robust topological feature over appreciable propagation distance with a near-unity skyrmion number. Our schema can also be further enhanced to explore more free-space propagation-invariant skyrmions[44] in the future.

In brief, a very promising direction is emerging for controlling more topological states beyond classical light in fiber-integrated metadevices, with ultrasmall volume and subwavelength feature to achieve ultra-high density information encoding. The metafiber design could also be extended for generating more diversified and topological quasiparticle states by superimposing sophisticated beams, for instance higher-order skyrmions, multiskyrmions, and 3D hopfions. Besides, the device size and processing time of metafiber can be further reduced through direct processing in the fiber facet[38,45], which is the trend

in the future. To enhance the compactness of the device, the polarizer can be replaced by performing polarization control in the all-fiber source or by introducing additional meta-lenses. Moreover, to realize a minimized device for tuning multi-textures, the QWP can also be optimized by using a rotating metasurface[46], or one-layer metasurface by combining phase change materials[47,48], two-dimensional materials[49], or liquid crystals[50], which can be the next step of future research. Therefore, the proposed of the metafiber skyrmion generator can strongly impact the development of next-generation information technologies. Given the versatility of meta-structures, various optical skyrmions can be further generated in this metafiber platform. The current fiber communication networks can be upgraded into metafiber networks where skyrmions are promising information carriers for larger-capacity information transfers. In addition, in contrast to conventional light waves, optical skyrmions can provide unconditional stability and resilience protected by their topologies against perturbations of disorders or turbulence in information channels, promising their usage of upgrading ultra-robust communication networks.

## Methods

### Metafiber fabrication

The metasurfaces were manufactured using the following processing procedure (the flow chart is shown in Supplementary Note 6). First, a 900 nm thick amorphous silicon is grown on a quartz substrate using Electron Beam Physical Vapor Deposition (EB-PVD). Then, the amorphous silicon layer is precisely etched using EBL to create the required structure, which includes a lattice period of 700 nm and antenna dimensions ranging from 250 nm to 550 nm. After etching, a chromium (Cr) layer is deposited on the amorphous silicon surface using Physical Vapor Deposition (PVD) technology, followed by a carefully designed lift-off process to remove excess chromium. Finally, the target metasurface is achieved through inductively coupled plasma (ICP) etching and the removal of the Cr layer. Meanwhile, the PSF (PM1550-HP) is attached to a fiber holder for stability, with the cross sections polished flat together. Then the glue is dripped between the fiber and the metasurface. For precise alignment of them, a homemade amplified light system is constructed to achieve a magnification of 100×. The thickness of the glue is precisely controlled by the micromotorized displacement stage. Finally, the proposed device can be obtained by curing the adhesive with a UV lamp.

### Experiment setup

A homemade optical measurement system is constructed to observe the performance of the metafiber skyrmion's generator, including an amplified light path and a coherent light path. The diagram of the experimental setup is shown in Supplementary Note 9. A homemade fiber laser operates at a wavelength of 1550 nm and outputs a power of 0−300 mW. Through a polarization beam splitter (PBS), lights with two polarizations are directed into the PSF of the light field amplification path and the coherent light path, respectively. The polarization controller attached to the laser output controls the intensity of light from the laser output to the two pathways. In the amplification path, one end output of PBS couples with metafiber to excite the skyrmions. As the lateral output spot of the skyrmions is on the sub-micron scale, and the size of a single pixel of the CCD (charge-coupled device, CinCam CMOS-1201) is 5.2 μm × 5.2 μm, a 100 × near-infrared objective lens is required for magnification. The objective lens is fixed on a motorized micrometer translation stage to move accurately. The polarizer is used to separate the light of different polarizations. In the coherent light path, the fiber is attached to a collimating mirror and light turns through a reflector. The size of light after collimation is a little bigger than the size of the metasurface. Then, the light from two light paths is merged through the beam splitter and interferes. The interference

light fields in $x$ and $y$ polarizations could be obtained by setting the directions of the two polarizers the same for $x$ and $y$, respectively.

### Characterizing topology of skyrmions

Topological properties of a skyrmionic configuration can be characterized by the skyrmion number[14]:

$$N_{sk} = \frac{1}{4\pi} \iint_\sigma \rho_{sk}\,dx\,dy \qquad (2)$$

where the skyrmion density is given by:

$$\rho_{sk} = \mathbf{S} \cdot \left( \frac{\partial \mathbf{S}}{\partial x} \times \frac{\partial \mathbf{S}}{\partial y} \right) \qquad (3)$$

where $\mathbf{S}(x,y) = [S_x(x,y), S_y(x,y), S_z(x,y)]$ ($S_x^2 + S_y^2 + S_z^2 = 1$) represents the 3D Stokes vector to construct a skyrmion and $\sigma$ is the region considered to confine the skyrmion. The skyrmion number is an integer counting how many times the vector $\mathbf{S}(x,y) = \mathbf{S}(r\cos\theta, r\sin\theta)$ wraps around the parametric unit sphere, i.e. the Poincaré sphere. For mapping to the unit sphere, the vector can be given by $\mathbf{S} = [\cos\alpha(\theta)\sin\beta(r), \sin\alpha(\theta)\sin\beta(r), \cos\beta(r)]$. The skyrmion number can be separated into two integers: the polarity, $p = \frac{1}{2}[\cos\beta(r)]_{r=0}^{r=r_\sigma}$, means that the vector direction is down (up) at center $r = 0$ and up (down) at the boundary $r \to r_\sigma$ for $p = 1$ ($p = -1$), and the vorticity, $m = \frac{1}{2\pi}[\alpha(\theta)]_{\theta=0}^{\theta=2\pi}$, controls distribution of the transverse field components. In the case of a helical distribution, an initial phase $\gamma$ should be added, $\alpha(\theta) = m\theta + \gamma$. For continuous regulation, the Jones matrix expression for a quarter-wave film rotated at $\theta$ is given by $\begin{bmatrix} 1 - i\cos 2\theta & -i\sin 2\theta \\ i\sin 2\theta & 1 + i\cos 2\theta \end{bmatrix}$. It is a unitary matrix, meaning that the QWP can be regarded as a rotation of the coordinate. Mathematically, the skyrmion number has been proven to be independent of the orthogonal coordinate system used[23]. From a topological point of view, this transformation is a smooth deformation and can be regarded as mapping the parametric sphere from different positions, which does not affect the topology as well as the skyrmion number.

## Data availability

All the technical details for producing the figures are provided in the supplementary information. The relevant data supporting this study are available from the corresponding authors Q.X. and Y.S. under request.

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

## Acknowledgements

Q.X. acknowledges the funding through National Natural Science Foundation of China (62122040, 62075113). Y.S. acknowledges the support of Nanyang Technological University Start Up Grant, Singapore Ministry of Education (MOE) AcRF Tier 1 grant (RG157/23), MoE AcRF Tier 1 Thematic grant (RT11/23), and Imperial-Nanyang Technological University Collaboration Fund (INCF-2024-007).

## Author contributions

T.H., Y.M., Y.S. and Q.X. conceived the idea. T.H. and Y.M. designed and numerically simulated the metasurface. T.H. performed the experiments with input from Y.M. and L.W. T.H., Y.M., Y.S. and Q.X. discussed the experimental design and analyzed the data. T.H., Y.M., and Y.S. prepared the manuscript with input from H.Z., N.M.C., D.L., P.Y. and Q.L. All authors discussed the results and contributed to revising the final manuscript. Q.X. and Y.S. supervised the project.

## Competing interests

The authors declare no competing interests.
