## [Transparent Peer Review file · Nature Communications]

Optical skyrmions from metafibers with subwavelength features

Corresponding Author: Professor Qirong Xiao

Version 0:

Reviewer comments:

Reviewer #1

(Remarks to the Author)

This manuscript reports on-fibre generation of optical skyrmions in a tight focus. This is a very interesting paper demonstrating high-quality optical skyrmions directly on single-mode optical fibres, which could achieve on-demand generation of complex structured light fields in the "plug and play" fashion. I recommend its publication more or less in its current form, but I do have a few minor comments for the authors to address:

1. Could the authors clarify how strong the tight focus is for observing the skyrmion distributions? Why is it necessary? Can it propagate just in free space in the paraxial limit? Did they add a metalens phase profile on the metasurface?
2. Following my above point, did the authors consider the depolarisation effect in a tight focus? This longitudinal electric field component might have impacted their skyrmion distributions? Please clarify.
3. I don't understand how the authors achieved imaging of subwavelength features of skyrmions beyond the diffraction limit. From the Supplementary experimental setup, it seems they have used a high numerical aperture lens for capturing the field with a magnification of 100, which suggests a numerical aperture of 1.4, so we expect a subwavelength resolution of around $\lambda/3$, how can they realise $\lambda/5.1$ or $\lambda/4.6$ in the experiment?
4. I wouldn't call the device "meta-skyrmion emitter", it's just a generation device, not an emitting device like the use of photoluminescence materials.
5. Some perspective on how to achieve diffraction-invariant propagation of skyrmions in free space would be helpful.

Reviewer #2

(Remarks to the Author)

This work authored by He et al designed a metasurface device to generate the high quality optical Skyrmion and Bimeron textures in the deep subwavelength scale. The traditional methods usually implemented the high NA focusing by microscopic objective to produce the field with subwavelength size. As mentioned by authors, this may lead to serious distortion of the resulting skyrmion patterns. This work well combines the technologies of optical-fiber and the metasurface to produce such an integrated system, which provides promising opportunities for the developments of optical communication and light signal processing using the interesting topological structured light such as the photonic skyrmion. Besides, the presented theoretical and experimental results are with good consistency. This manuscript has clear writing, which is good to the reader who is interesting to this topic. There are several problems require to be addressed at this stage, before I can recommend to accept this manuscript in Nature Communications.

The longitudinal field cannot be neglected under the employment of the high NA focusing, however, the relevant analysis is absent in the current manuscript. As a consequence of that effect, I am wondering if the photonic skyrmionic structure can be

maintained. I think that this ignored z-component can affect the measured skyrmion textures, and I wonder, why it can be omitted in this configuration.

The title of this manuscript is related to the concept of subwavelength. However, the reader might concern about how previous researches generate the subwavelength structured light. Therefore, I recommend to review this field and make relevant revision on the introduction. I believe that adding those reviews and relevant references can strongly clarify the novelty of implementing optical fiber device, rather than the conventional methods.

One relevant comment related to the subwavelength: I cannot clearly observe the subwavelength feature size of the photonic skyrmion both from the figures and the text. I suggest to show this subwavelength feature size in the figure clearly at proper position. For example, what is the full width at half maximum of the photonic skyrmion? What is the width of the whole field? How to measure this subwavelength feature experimentally. I think that these details are essential when concerning the subwavelength.

The manuscript has provided the details about fiber fabrication, the characterization of these optical quasi-particles. However, there are still some crucial techniques that have not been introduced. The procedures of getting the experimentally measured phases of nano fields are absent in this manuscript, corresponding to the insets of bottom panels in Fig. 4b. I think this technique is important for the reproduction by the following studies. In addition, the QWP was used to adjust the output state of skyrmion. This process, in some extent, is low integration, which may enlarge the device volume. I suggest the authors to discuss the solutions to improve the integration level in the outlook part.

Reviewer #3

(Remarks to the Author)

In this paper He and coworkers design a metasurface that will create skyrmions when input with the correct polarisation, and then fabricate this on the face of a fibre. This makes the "source" of the skyrmions very compact and I think it is the first reported use of a metasurface for skyrmions (at least it is the first where the term skyrmion is mentioned). They then show how this can be used in a controllable fashion, and claim sub-wavelength features. The paper is very well written and beautifully illustrated, with very convincing theoretical and experimental data that are in support of one another and (mostly) the main claims. The importance of the work is that it opens a new way to create and deliver these highly topical structured light fields. I have a few suggestions for improvement:

* the introduction is good but there are parts that could be better. For instance, it could be given a broader context of structured light and vectorial complex fields, and the references to resilience are not correct in my opinion: [21] argues that they are NOT resilient in some cases and is purely theoretical in nature, [22] is a creation tool so cited incorrectly, [23] is indeed resilience of concurrence of vectorial fields which is related but does not imply that skyrmions are resilient, [24] has no perturbation so resilience is meaningless and [25] has not been published and is still questionable. As far as I am aware, the first experimental report on resilience to a perturbation is ref [51] (and an associated arxiv on noise resilience that came out before [25]) which is unfairly only cited in the methods as a 'tool', whereas I think it should also be moved to the intro. All the references used are good, but the authors should unpack them in a way that does not mislead the reader. In fact, we should be clear that what matters is if the perturbation is map preserving or not. In photonics it may not always be.

* Plot S6 is not correct. If you look carefully at the theory you will see that skyrmions are defined for maps from spheres to spheres, which implies an integer value. Once the value is non-integer the map is not from spheres to spheres, so no longer skyrmionic. So rather than the 'skymre number' on the vertical axis the authors should call this 'PS coverage', since this is what is actually calculated and is indeed variable continuously. It is interesting to ask what the maps are when the integer value is broken - a topic for another day.

* I am not convinced by the results of Fig 4e and the associated SI. The authors basically argue that the diffraction limit holds for "other" measurements but here the S_z can be resolved even better. First, S_z is not so common a term in Stokes analysis so may be unclear to some. Second, Stokes data is by definition spatial- camera images - so if the camera cannot beat the diffraction limit, how can something derived from it? The authors would have to show spatially resolved amplitude features to convince me. And if the results are correct (happy to hear the counterargument), what is the nature of this enhancement? I could not clearly discern this.

* I suggest that the authors carefully look at the number of self-citations. We all think our own work is important but in such a high impact journal the work should appeal to a wide audience. It does not aid that argument of the reference suggest that only one group are working in the field.

Version 1:

Reviewer comments:

Reviewer #1

(Remarks to the Author)

I am happy for all the revisions that the authors have made, and I can endorse its publication now.

Reviewer #2

(Remarks to the Author)

Many thanks for sending me the revised manuscript, which has been significantly improved and clarified, particularly regarding the concept of the subwavelength-scale skyrmions. The authors defined the subwavelength-scale skyrmions using the photonic polarization rather than the intensity of light. This leads to a deep-subwavelength feature size of the skyrmions (represented by the Stokes parameters), which indeed is not reported before and difficult to be achieved to my knowledge. On the other hand, it is impressive that the authors have carefully addressed all the reviewers' comments and suggestions and added relevant results in the response file and the supplementary for further clarifications. At this stage, I have no other comments on the manuscript and in my opinion the revised manuscript can be accepted by Nature Communications in its current form.

Reviewer #3

(Remarks to the Author)

This is a much better version that still has the impact of the original but more clearly conveyed and better representative of the complementary work.

Detailed Responses to the Reviewers

We sincerely appreciate all the constructive comments raised by the reviewers. We have carefully revised our manuscript accordingly with the detailed responses as the following.

Response to Reviewer #1

Comment: This manuscript reports on-fiber generation of optical skyrmions in a tight focus. This is a very interesting paper demonstrating high-quality optical skyrmions directly on single-mode optical fibres, which could achieve on-demand generation of complex structured light fields in the "plug and play" fashion. I recommend its publication more or less in its current form, but I do have a few minor comments for the authors to address:

Response:

Many thanks for your positive feedback and insightful comments regarding our manuscript on the on-fiber generation of high-quality optical skyrmions. We appreciate your recognition of the significance of our work and your recommendation for publication. We have made the following revisions to enhance the clarity and quality of the manuscript. Thank you again for your valuable insights.

Comment 1: Could the authors clarify how strong the tight focus is for observing the skyrmion distributions? Why is it necessary? Can it propagate just in free space in the paraxial limit? Did they add a metalens phase profile on the metasurface?

Response 1:

Many thanks for your comment. We apologize for the missing statement of a clear statement of the focusing condition. We have clarified this in our revised version: we utilized a tightly focused beam with a numerical aperture (NA) of **0.8** to observe the skyrmion distributions in our manuscript.

The tight focus with high NA is not necessary for general optical skyrmion generation, but it is necessary to reach deep-subwavelength features of optical skyrmions (also detailed in Response 3) in this manuscript. We can also design using not-high NA (loose focus) to generate a paraxial weakly focused skyrmionic beam. Nevertheless, we believe the high-NA focused skyrmions with subwavelength feature are **more challenging to implement, as the subwavelength-level Stokes skyrmions were not experimentally reported before**. Besides, it holds great potential for extraordinary applications such as local high-density information multiplexing and storage, as well as super-resolution imaging and sensing.

Regarding the question of the metalens phase profile – Yes, we indeed added a metalens phase profile on the metasurface to compensate for the fiber source propagation phase. In the metafiber-tip design, there is a distance of 350 μm between the PSF facet and the metasurface for beam expansion. The beam will diffuse, and the phase distribution will be spherical. Thus, we compensate it by subtracting this phase in our metasurface design to eliminate its effect. We have also added a clearer description in revised manuscript (around line 131-132) and Supplementary Notes 1.

Comment 2: Following my above point, did the authors consider the depolarisation effect in a tight focus? This longitudinal electric field component might have impacted their skyrmion distributions? Please clarify.

Response 2:

Thank you for the rigorous consideration. Yes, we considered the depolarisation effect and the longitudinal electric field component throughout the entire process. As **we used full-vector FDTD simulation throughout the entire process (neither scalar electric field nor paraxial assumptions were applied), the depolarisation effect and the longitudinal electric field components are all rigorously and comprehensively considered** in all the calculation results presented in this manuscript.

The longitudinal electric field component may have a slight impact on the shape of the transverse electric field distributions. However, **it barely effects the quality of the constructed Stokes skyrmions reported in this manuscript.**

The reason is that the standard Stokes skyrmions reported throughout this manuscript are defined using the transverse electric field components at the central confinement region (E_x and E_y ; as detailed in Supplementary Note 3 and Methods sections; with similar concept in Ref. [ACS Photonics 9, 296-303 (2022)] and Ref. [Nature Photonics 18, 15-25 (2024)]). The presence of longitudinal field components will affect the spatial 3D polarization texture, but will not affect the transverse polarization profiles and the topological features of the Stokes skyrmions. It is unlike the other skyrmions constructed by spin (Ref. [Nature Physics 15, 650-654 (2019)]) or electric field (Ref. [Nature Communications 12, 5891 (2021)]), which are largely affected by the longitudinal electric field components.

Fig. R1 The Stokes Poincaré spheres with corresponding skyrmion numbers (upper) and vector distributions (lower) of skyrmions over increasing propagation distances (z) after the fiber facet within the depth of focus of Bessel beams.

In order to verify the effect of E_z on our constructed skyrmion, we also added the analysis of the generated skyrmion at different propagation distances within Bessel beam's nondiffracting distance (shown in Fig. R1 above). From the results, we can see the polarization distributions as well as the topological features are almost unaffected and stable during propagation after the fiber facet, demonstrating an excellent property of our Stokes skyrmion (also further clarified in Supplementary Note 5).

In short, the presence of longitudinal electric component E_z will only impact the spatial polarization distribution, but will not highly disrupt the transverse polarization distributions and the topological features of the constructed Stokes skyrmions reported in this manuscript.

Comment 3: I don't understand how the authors achieved imaging of subwavelength features of skyrmions beyond the diffraction limit. From the Supplementary experimental setup, it seems they have used a high numerical aperture lens for capturing the field with a magnification of 100, which suggests a numerical aperture of 1.4, so we expect a subwavelength resolution of around wavelength/3, how can they realise wavelength over 5.1 or 4.6 in the experiment?

Response 3:

Thanks for the insight question and we apologize for the ambiguous explanation on this regard. Indeed, we agree with the referee that conventionally, using $NA=1.4$ and a magnification of $100\times$ will expect a resolution of $\sim\lambda/3$. However, this refers to the resolution of a scalar intensity-based light spot distribution, not for polarization variation in Stokes vector pattern. In the revised version, we have made the corresponding clarifications (also explained as the following) on how to reach the subwavelength features of skyrmions.

Definition of subwavelength features: The 'subwavelength features' raised in this manuscript refers to the 'polarization features' (re-clarified in section Experimental results on Page 9) instead of the 'light intensity subwavelength features' used in conventional imaging systems. **The direction of the electromagnetic, which defines the polarization and local photonic spin state of light, is not directly subject to conventional light intensity distribution, and can vary at much smaller scales** [e.g. as Ref. Nature Physics 15, 650-654 (2019)]. In our schema, the subwavelength features of the skyrmion polarization patterns are characterized by the variation of the normalized S_z component of the polarization Stokes vector (S_x, S_y, S_z) which is subjected to the 2D transverse electric profiles.

We also plot the radial variations of the normalized intensity of transverse field (I_{RCP} and I_{LCP}) and the longitudinal stokes parameter $|S_z|$ from our simulations in Fig. R2 to give a clearer illustration. From the figure, we can see the polarization variation is not directly subject to the intensity spot and changes on thinner spatial scales (simulated: $\sim\lambda/5.6$, experiment: $\sim\lambda/5.1$). It can be highly beneficial for precision metrology [Ref. Advanced Science, 10, 2205249 (2023)], particle manipulation, super-resolution imaging and etc.

Fig. R2 The radial variations of the normalized intensity of transverse field (I_{LCP} and I_{RCP}) and the Stokes parameter $|S_z|$

Experimental acquisition: The subwavelength polarization feature **is not directly observed by camera, but is derived from the measured electric fields under orthogonally polarized states in experiments**, that is, S_z can be obtained. The process is as follows: we experimentally captured the interference intensity profiles in the x - and y - polarized states, which in turn recover the complex amplitudes (FWHM $\sim \lambda/2.5$), and then calculate the polarization Stokes parameters to obtain the S_z components (simulated FWHM $\sim \lambda/5.6$; experimentally verified FWHM $\sim \lambda/5.1$). We use the Bessel beam as basis scalar beams, which can realize a smaller FWHM than Gaussian beams with same NAs (Similar to Ref. [Light: Science & Applications **6**, e16259-e16259 (2017)], which generates the Bessel beams as small as $\sim \lambda/3$).

In brief, we use the Bessel beams as basis scalar beams and realize the light intensity-profile-based FWHMs of $\sim \lambda/2.5$, and we use a high numerical aperture (NA) lens for capturing the Bessel fields in our experiment. Nevertheless, its theoretically simulated and experimentally verified ‘polarization subwavelength features’ (around $\sim \lambda/5$) can be much smaller to diffraction limit.

Comment 4: I wouldn't call the device "meta-skyrmion emitter", it's just a generation device, not an emitting device like the use of photoluminescence materials.

Response 4:

We fully agree with the reviewer and the suggestion has been addressed.

We have revised our manuscript by re-calling it as “meta-skyrmion generators” to describe more accurately the function and nature of the device (in line 277, and etc.).

Comment 5: Some perspective on how to achieve diffraction-invariant propagation of skyrmions in free space would be helpful.

Response 5:

Thank you for the constructive comments. We fully agree the significance of diffraction-invariant propagation of skyrmions in free space. For the Stokes skyrmions reported in this manuscript, the quasi-Bessel beams are used as the basis beams, and a quasi-diffractionless invariant propagation of Stokes parameters (realize diffraction-invariant 2D polarization textures) have already achieved

within the depth of focus $\frac{D}{2 \tan[\arcsin(NA)]} \sim 20 \mu\text{m}$, where D is the diameter of the metasurface. To

generate diffraction-invariant skyrmions with constant 3D polarization textures from the image plane of the metasurface, it is imperative to manipulate the complex amplitude of the orthogonal polarization components to match the profiles with the free-space eigenmodes (LG, BG...). In our work we only modulate the phase, but it can be extended to realize robust propagation-invariant skyrmions in free space in future. We have added corresponding discussions in the conclusion section (“Our schema can also be further enhanced to explore more free-space propagation-invariant skyrmions [e.g. Ref. Optics Express 31, 15289-15300 (2023)] in the future.”).

Response to Reviewer #2

Comment: This work authored by He et al designed a metasurface device to generate the high quality optical Skyrmion and Bimeron textures in the deep subwavelength scale. The traditional methods usually implemented the high NA focusing by microscopic objective to produce the field with subwavelength size. As mentioned by authors, this may lead to serious distortion of the resulting skyrmion patterns. This work well combines the technologies of optical-fiber and the metasurface to produce such an integrated system, which provides promising opportunities for the developments of optical communication and light signal processing using the interesting topological structured light such as the photonic skyrmion. Besides, the presented theoretical and experimental results are with good consistency. This manuscript has clear writing, which is good to the reader who is interesting to this topic. There are several problems require to be addressed at this stage, before I can recommend to accept this manuscript in Nature Communications.

Response:

We sincerely appreciate the constructive comments raised by the reviewer. We have carefully revised our manuscript accordingly to further enhance with corresponding clarifications on the definitions and discussions on the polarization subwavelength features and related contents with highlighted significance and novelty. Detailed revisions responses are as follows.

Comment 1: The longitudinal field cannot be neglected under the employment of the high NA focusing, however, the relevant analysis is absent in the current manuscript. As a consequence of that effect, I am wondering if the photonic skyrmionic structure can be maintained. I think that this ignored z-component can effect the measured skyrmion textures, and I wonder, why it can be omitted in this configuration.

Response 1:

Many thanks for the rigorous consideration. The longitudinal electric field E_z was not ignored in the metasurface simulation and Stokes skyrmion construction. And the presence of E_z does not effect the topological features (i.e. skyrmion number or texture of realized Stokes skyrmions).

The longitudinal component (E_z) is already under rigorous consideration in our manuscript.

As we applied full-vector FDTD simulations in the whole process, the effect of non-negligible z-component E_z is already rigorously considered in our schema. No scalar electric field assumptions or paraxial assumptions were used and the E_z component was not omitted.

Besides, regarding the construction criteria of the optical skyrmions here, the skyrmion indeed has many types, such as spin, stokes and electric field. In the manuscript, we focused on Stokes skyrmions (Ref. [Phys. Rev. A 102, 053513 (2020)], Ref. [Phys. Rev. Res. 3, 023055 (2021)]) thanks to its great flexibility in design and good potential for detection after propagation to far field. Compared to other spin (Ref. [Nat. Phys. 15, 650–654 (2019)]) and electric field skyrmions (Ref. [Nature Communications 12, 5891 (2021)]), **the Stokes skyrmions we have studied are solely constructed by the transverse components (E_x and E_y) and their corresponding Stokes vector (S_x, S_y, S_z) are derived from these transverse components.**

For the effect of E_z : The standard definition of the Stokes parameters does not include the longitudinal component, therefore is not considered in the Stokes skyrmion construction (Supplementary Note 3), but its effect has already been considered in the results. The longitudinal electric field component E_z has an impact on 3D spatial polarization textures of the generated structured light field, but **it does barely impact the transverse 2D polarization textures as well as the topological skyrmion number of the constructed Stokes skyrmions reported in this manuscript.**

We have also added the additional simulations on the construction of Stokes skyrmion over different distances in the depth of focus of quasi-Bessel beams to analyze the effect of E_z . The intensities of the electric fields under orthogonal polarizations and the Stokes vector distributions are illustrated in Fig. R3 below. The Poincaré spheres and corresponding skyrmion numbers are shown in Fig. R4 below. From the simulation results, we can clearly see that the electric fields and vector distributions are stable with rare variation due to the E_z , and the coverages of the Poincaré sphere, i.e. the skyrmion number are kept nearly unchanged (also added into revised Supplementary Note 5). It can be seen that the Stokes skyrmions we generated not only propagate to the far field but also maintain a very stable and unchanged topological features over a nondiffracting distance.

Fig. R3 The electric fields in x -pol (the first row) and y -pol (the second row) and the vector distributions (the third row) for different propagation distances (different columns). Added as Supplementary Fig. S7.

Fig. R4 The Poincaré spheres of skyrmions over increasing propagation distances (z). Added as Supplementary Fig. S8.

Comment 2: The title of this manuscript is related to the concept of subwavelength. However, the reader might concern about how previous researches generate the subwavelength structured light. Therefore, I recommend to review this field and make relevant revision on the introduction. I believe that adding those reviews and relevant references can strongly clarify the novelty of implementing optical fiber device, rather than the conventional methods.

Response 2:

We appreciate the constructive comments raised by the reviewer. As suggested by the reviewer, we have further revised the field of the subwavelength structured light and have added some relevant and typical literatures on the revised introduction (around line 50 ~ line 52).

Previous researches significantly advanced the field of subwavelength structured light. We believe our Stokes skyrmions with subwavelength polarization patterns represent a meaningful and miniaturized step forward in this area.

The following references are added to the revised manuscript:

- Ref. 28 Zhang, X., Liu, G., Hu, Y., Lin, H., Zeng, Z., Zhang, X. *et al.* Photonic spin-orbit coupling induced by deep-subwavelength structured light. *Physical Review A* **109**, 023522 (2024).
- Ref. 29 Zhang, X., Hu, Y., Zhang, X., Li, Z., Chen, Z. & Fu, S. On-Demand Subwavelength-Scale Light Sculpting Using Nanometric Holograms. *Laser & Photonics Reviews* **17**, 2300527 (2023).
- Ref. 30 Davis, T. J., Janoschka, D., Dreher, P., Frank, B., Meyer zu Heringdorf, F.-J. & Giessen, H. Ultrafast vector imaging of plasmonic skyrmion dynamics with deep subwavelength resolution. *Science* **368**, eaba6415 (2020).

We have also carefully revised it in the manuscript:

50 electromagnetic interference. Compared to current systems that generate subwavelength structured
 51 light²⁸⁻³⁰, which typically have large optics, fiber optic exhibit tremendous potential to realize
 52 flexible optical device integration with decent miniaturization, flexibility, and lightweight. Driven

Comment 3: One relevant comment related to the subwavelength: I cannot clearly observe the subwavelength feature size of the photonic skyrmion both from the figures and the text. I suggest to show this subwavelength feature size in the figure clearly at proper position. For example, what is the full width at half maximum of the photonic skyrmion? What is the width of the whole field? How to measure this subwavelength feature experimentally. I think that these details are essential when concerning the subwavelength.

Response 3:

Many thanks for your considerate comment.

In our manuscript, the subwavelength feature is defined as the variation of the S_z component in the polarization Stokes component (S_x, S_y, S_z); i.e. **the subwavelength features here are the fine polarization textures, which differ from the subwavelength features in the light intensity spots in conventional imaging systems, and can vary in much finer scales.** We apologize for the potential confusion caused and we have made further clarifications to address it. We also revised the manuscript, further clarifying the polarization subwavelength features of Stokes skyrmions.

276 Furthermore, subwavelength feature of the polarization Stokes parameters are also realized in
 277 our proposed meta-skyrmion generators. The direction of the electromagnetic, which defines the
 278 polarization and local photonic spin state of light, is not directly subject to conventional light
 279 intensity spot, and can vary at much smaller scales⁹. The variation of longitudinal stokes parameter
 280 S_z with spatial position illustrates how the polarization state of light changes throughout the space.

We also plot the radial variations of the normalized intensity of transverse field (I_{RCP} and I_{LCP}) and the longitudinal stokes parameter $|S_z|$ from our simulations in Fig. R5 to give a clearer illustration. The polarization inversion width is defined as the distance between two points where the normalized Stokes parameter S_z changes sign and reaches half its maximum value, indicating the scale of the coverage of half the Poincaré sphere (from the upper-half parallel to the lower-half parallel). From the figure, we can see the polarization variation is not directly subject to the intensity spot and changes on thinner spatial scales ($\sim \lambda/5.6$).

Fig. R5 The radial variations of the normalized intensity of transverse field (I_{LCP} and I_{RCP}) and the stokes parameter $|S_z|$

Thus, the subwavelength polarization feature of Stokes skyrmion mentioned in our manuscript are shown below (also illustrated as Fig. 4e in the revised manuscript). It can be seen that both skyrmion and bimeron reverse the polarization states with fine polarization texture features of $\lambda/5.1$ and $\lambda/4.6$ respectively, which is not limited by light-intensity-based size.

Fig. R6: The normalized Stokes parameter and its absolute value versus x of skyrmion (left) and bimeron (right) states, with the vector distributions of experiment results inserted.

Besides, we also added the illustration of the electric field distributions on two orthogonally polarizations in Fig. R7 to show the size of skyrmions. The width of the plotting window is $2 \mu\text{m}$. The Stokes skyrmion is constructed by the J_0 in x -pol and J_1 in y -pol in our manuscript. Thus we use the FWHMs of two basis scalar beams to characterize the size of skyrmion. Herein, the FWHM of the J_0 is defined as the distance between two points at half of the maxima intensity of the center bright spot. Similarly, the FWHM of J_1 is defined as twice the distance from the dark spot center to the point at its closest ring with the half-maximal intensity. In general, BB's FWHM is characterized by the average of the FWHMs of the fast and slow axes. (Ref. [Light: Science & Applications **6**, e16259-e16259 (2017)]). The simulated profiles are plotted in the left panels of Fig. R7, and the experimental profiles are plotted in right panels, with the J_0 in the upper and J_1 in the lower. The experimental FWHMs of J_0 and J_1 are $0.69 \mu\text{m}$ and $0.58 \mu\text{m}$ respectively, showing great consistence with the simulated FWHMs of $0.67 \mu\text{m}$ and $0.57 \mu\text{m}$ respectively. The little discrepancies are analyzed in the revised Supplementary Note 13.

Fig. R7 The intensity profiles at the transverse plane for (a) the simulation for x -pol, (b) the experiments for x -pol, (c) simulation for y -pol, (d) experiments for y -pol.

For the comment on **experimental realization**, we can collect the intensity distributions of the interference optical fields and plane waves under orthogonal polarization states through experimental measurements by CCD. By using the process elaborated in Response 4, we can retrieve the complex amplitude distributions. Through the calculation formula clarified in Supplementary Note 3, we can obtain the polarization distributions characterized by the Stokes parameters, which allows for the determination of the S_z component. The polarization inversion width of the photonic skyrmion is defined as the distance in which S_z varies from 0.5 to -0.5, thus showing the scale in which half the Poincaré sphere is covered (from the upper-half parallel to the lower-half parallel). Therefore, we can obtain this polarization inversion width as small as $\sim \lambda/5$.

Comment 4: The manuscript has provided the details about fiber fabrication, the characterization of these optical quasi-particles. However, there are still some crucial techniques that have not been introduced. The procedures of getting the experimentally measured phases of nano fields are absent in this manuscript, corresponding to the insets of bottom panels in Fig. 4b. I think this technique is important for the reproduction by the following studies. In addition, the QWP was used to adjust the output state of skyrmion. This process, in some extent, is low integration, which may enlarge the device volume. I suggest the authors to discuss the solutions to improve the integration level in the outlook part.

Response 4:

We appreciate the valuable comment, and this suggestion has been carefully addressed.

We have added a more detailed explanation regarding the procedures for phase characterizations from the experimentally measured intensity profiles. Additionally, we also revised Supplementary Note 8 as well as Supplementary Note 9 to make a clearer and more precise descriptions on experimental setup and the phase recovery technique.

From the experimental setup shown in Fig. S15, we could obtain the intensity profiles of the interference waves under x and y polarizations. The complex amplitude of interference wave can be obtained using the procedure illustrated in Fig. R8.

The procedure of recovering the complex amplitude of the interference fields is illustrated in Fig. R8. Firstly, the intensities of interference fields (first left column) in x (upper row) and y (lower row) polarizations are collected experimentally. Then, the interference intensity is Fourier transformed. The Fourier transformed image (second column) consists of three main parts: the zeroth-order containing the background information, the first order +1, and the conjugate order -1, the last two parts containing the phase information. Due to the completely off-axis interference mode, the zeroth order, +1- and -1- orders completely separated in the spectrum of the interferogram. Subsequently, by spatially filtering the first order (third column) and performing inverse Fourier transform, we can obtain the complex amplitudes of the interference profiles ($E_{\text{interference}} = E_{\text{object}} \times E_{\text{plane}}^*$), shown in right-most column.

Fig. R8 The process of recovering the complex amplitude of interference fields under (a-d) x polarization and (e-h) y polarization. (As Supplementary Fig. S16 in revised SI)

Meanwhile, the intensity profiles of the collimated wave are also obtained in experiments (the phase profiles of which are planar). The complex amplitude profiles of the object wave can be derived by dividing the complex amplitude of the interference wave by the conjugated reference plane wave ($E_{\text{object}} = E_{\text{interference}} / E_{\text{plane}}^*$). The figures are also shown in the added Fig. S17 and Fig. S18 in the Supplementary Information.

Fig. R9 The process of Complex amplitude recovery in x polarization in experiments. The complex amplitude of the object wave (third column) is derived by dividing the interference complex amplitude field (first column) by the conjugate complex amplitude profile of the coherent plane beam (second column). (As Supplementary Fig. S17 in revised SI)

For the comment on the QWP, we need to note that the QWP used in our work is primarily for result validation with easy experimental implementation (proof of concept) to show the tunable/configurable skyrmion texture by the QWP. Therefore, the QWP would be in general not necessary for shaping topological Stokes skyrmion. For realizing one specific texture like skyrmion or bimeron, alternatively changing the polarization basis or the antennas size is enough to realize. We have added this perspective to our revised manuscript detailed below.

For regulating multi-textures on one device, like tuning the two different skyrmions and/or bimerons, a rotating metasurfaces (such as Ref. [Advanced Optical Materials 10, 2102166 (2022)]) or one-layer metasurface by combining phase change materials (such as VO₂ in Ref. [Optics Express 32, 5862-5873 (2024)] or GST in Ref. [Laser & Photonics Reviews 10, 986-994 (2016)]), two-dimensional materials (Ref. [Nature Reviews Materials 8, 498-517 (2023)]), or liquid crystals (Ref. [Nano Letters 21, 4554-4562 (2021)]), the tunability between different quasiparticle states can be further enhanced. We have thus added relevant discussions on how to further enhance integration incorporating QWP or reconfigurable devices accordingly in revised manuscript (around lines 325-328).

Response to Reviewer #3

Comment: In this paper He and coworkers design a metasurface that will create skyrmions when input with the correct polarisation, and then fabricate this on the face of a fibre. This makes the "source" of the skyrmions very compact and I think it is the first reported use of a metasurface for skyrmions (at least it is the first where the term skyrmion is mentioned). They then show how this can be used in a controllable fashion, and claim sub-wavelength features. The paper is very well written and beautifully illustrated, with very convincing theoretical and experimental data that are in support of one another and (mostly) the main claims. The importance of the work is that it opens a new way to create and deliver these highly topical structured light fields. I have a few suggestions for improvement:

Response:

We would like to sincerely thank the reviewer for the detailed review and constructive comments on further improving the quality of our manuscript. We have carefully revised our manuscript accordingly with detailed responses as the following.

Comment 1: the introduction is good but there are parts that could be better. For instance, it could be given a broader context of structured light and vectorial complex fields, and the references to resilience are not correct in my opinion: 21-25. [21] argues that they are NOT resilient in some cases and is purely theoretical in nature, [22] is a creation tool so cited incorrectly, [23] is indeed resilience of concurrence of vectorial fields which is related but does not imply that skyrmions are resilient, [24] has no perturbation so resilience is meaningless and [25] has not been published and is still questionable. As far as I am aware, the first experimental report on resilience to a perturbation is ref [51] (and an associated arxiv on noise resilience that came out before [25]) which is unfairly only cited in the methods as a 'tool', whereas I think it should also be moved to the intro. All the references used are good, but the authors should unpack them in a way that does not mislead the reader. In fact, we should be clear that what matters is if the perturbation is map preserving or not. In photonics it may not always be.

Response 1:

Many thanks for the constructive comment. We fully agree with the reviewer that the introduction can be improved with clearer statements of prior works.

The suggestion has been fully taken. We have added several typical articles about structured light and vectorial complex fields into the introduction. In addition, we have revisited the literatures and provided a clearer categorization along with more accurate discussions.

For the references the referee mentioned:

Ref. [21] [Physical Review Letters 129, 267401 (2022)], indeed, is not reflecting resilience, while it is an impressive work theoretically studying state transition of skyrmions in disorder media, providing good platform study resilience in the future.

Ref. [22] [Optica 9, 187-196 (2022)] can be a nonlinear transformation of general vector beams including optical skyrmions, which studied the structural stability of skyrmion upon special nonlinear conversions. We have revised our manuscript accordingly.

Ref. [23] [Nature Photonics 16, 538-546 (2022)] is a general theoretical model, indeed not specific to skyrmions, and includes resilience. It could be potentially extended to study skyrmionic beam stability. We agree with the referee and have revised the related discussions.

Ref. [24] [Appl. Phys. Rev. 11, 031411 (2024)] describes how the electromagnetic skyrmionic pulse can self-repair its structure in free space propagation. We agree with the referee that there is not perturbation during this process, nevertheless, we anticipate this work can be extendable to study resilience with complex media perturbation in the future. As the skyrmion propagates, its skyrmion number will be repaired and increased.

For Ref. [25] [arXiv:2403.07837 (2024)], we agree with the referee that this Ref is still questionable. Therefore, we have removed this Ref in our revised manuscript.

For Ref. [51] [Nature Photonics 18, 258-266 (2024)], we have re-organized its location. As it includes resilience contents, we briefly mentioned it in the introduction but with more accurate modified description.

We have thus re-organized the literatures and carefully revised the corresponding discussions in introduction section (around line 39 ~ line 41) for enhanced preciseness and clarity.

39 spin-orbit coupling and tunable topological textures¹⁷⁻¹⁹ showcase unique features such as
40 topological stability²⁰, topological state transition in disorder media²¹, structural stability in
41 nonlinear conversion²² or resilience in free-space propagation^{23,24}, exhibiting great potential as

Comment 2: Plot S6 is not correct. If you look carefully at the theory you will see that skyrmions are defined for maps from spheres to spheres, which implies an integer value. Once the value is non-integer the map is not from spheres to spheres, so no longer skyrmionic. So rather than the 'skymre number' on the vertical axis the authors should call this 'PS coverage', since this is what is actually calculated and is indeed variable continuously. It is interesting to ask what the maps are when the integer value is broken - a topic for another day.

Response 2:

We appreciate the reviewer for raising this constructive comment. We have changed the vertical axis of Fig. S6 to 'PS coverage' accordingly, which is indeed more suitable and accurate than 'skymre number'.

Fig. R10 The PS coverage as a function of the polarization angle θ .

Indeed, according to the definition, a skyrmion number of a perfect skyrmion theoretically must be an integer. However, in experimental implementations, the usage of non-integer skyrmion number also exists; for example, many previous foundational references (like Ref. [Science 361, 993-996 (2018)], Ref. [Science 368, eaba6415 (2020)], Ref. [Optica 11, 769-775 (2024)]) also used a **non-integer skyrmion number to characterize the similarity to the perfect skyrmion, as well as the quality of resemblance of the experimentally generated beam with perfect skyrmions**. We have also added a clear explanation about non-integer skyrmion number in manuscript (lines 192 - 194). Some exemplary non-integer skyrmion numbers used in prior references to evaluate the quality of generated structured light are attached below:

[3 figures redacted]

Fig. R11 The exemplary non-integer skyrmion numbers used in prior references

Comment 3: I am not convinced by the results of Fig 4e and the associated SI. The authors basically argue that the diffraction limit holds for "other" measurements but here the S_z can be resolved even better. First, S_z is not so common a term in Stokes analysis so may be unclear to some. Second, Stokes data is by definition spatial- camera images - so if the camera cannot beat the diffraction limit, how can something derived from it? The authors would have to show spatially resolved amplitude features to convince me. And if the results are correct (happy to hear the counterargument), what is the nature of this enhancement? I could not clearly discern this.

Response 3:

Many thanks for the constructive comment. We realized that there was an inconsistent use of symbols for the Stokes parameters (S_1, S_2, S_3 and S_x, S_y, S_z that referred to the same thing) in original manuscript. We would like to apologize for the confusion caused and greatly appreciate the reviewer for pointing it out. We have revised and unified the symbol annotations in the revised manuscript accordingly: unifying them to $S_x, S_y,$ and S_z .

(1) Regarding the meaning and physical term of S_z : The Stokes parameters (S_x, S_y, S_z) characterize the polarization state of an optical field, and S_z is indeed a commonly used physical parameter which indicates the degree of right-circular polarization. We added Fig. R12 to provide a more intuitive explanation on S_z .

The left panel shows a 2D distribution of polarization ellipsoids, and the right panel is 1D plot of S_z corresponding to the black line in the left panel. The red color represents left-handedness, and the blue color represents right handedness. The sign of S_z indicates the handedness of the polarization (right or left), and its magnitude reflects the degree of circular polarization of an elliptical state. When $|S_z| = 1$, it represents pure circularly polarized light; when $|S_z| = 0$, it corresponds to linearly polarized light. Values of ($|S_z|$) between 0 and 1 indicate elliptically polarized light. Thus, the variation of S_z with spatial position illustrates how the polarization state of light changes throughout the space.

Fig. R12 Left: A 2D polarization ellipsoids profiles. Right: The 1D plot of S_z corresponding to the black line in left panel. (Red color represents RCP, and blue color represents LCP)

(2) Regarding the subwavelength features, we have further clarified its concept and definition in our revised manuscript on the subwavelength ‘polarization texture’ of our generated Stokes skyrmions. A visual illustration is shown below (as Fig. R13) with curves of the radial variations of the normalized intensity of the transverse fields and the corresponding stokes parameter $|S_z|$. S_z is defined through the intensity difference between LCP and RCP components, so if one of them (RCP) drops to the half peak value while the other one (LCP) increases within the same length, the resulting S_z will change according to the combined variation of both profiles at the same time, so that it can feature smaller FWHM than the constituent intensity profiles. For sake of clarification, we have plotted the intensity profiles of a gaussian beam (RCP/H) and LG vortex (LCP/V), and the corresponding normalized S_z . As seen in Fig. R14 (left picture), the FWHM of S_z , measured as the width when $S_z = 0$, is significantly smaller than the FWHM of the scalar component (RCP). If we measure the FWHM of S_z , as the distance of when S_z goes from 0.5 to -0.5 (upper-half parallel in the Poincaré sphere to lower-half parallel), is also smaller than the FWHM of the scalar components. Therefore, the variation of S_z (distinctive concept to the intensity profile-based diffraction limit-spots) is intimately related to the combined variation of both orthogonal components at the same time, which makes its FWHM that can go beyond the FWHM of the scalar components.

Fig. R14 The radial variations of the intensity of transverse field (Left: I_{LCP} and I_{RCP} ; Right: I_H and I_V) and the stokes parameter S_z ($y = 0$).

We have also depicted the radial variation of transverse intensity (I_{RCP} and I_{LCP}) and the $|S_z|$ distribution of the simulation results from our proposed device in Fig. R15. The horizontal coordinate is x/λ , and the distance in which we cover half the Poincaré sphere (upper-half parallel to lower-half parallel; S_z from 0.5 to -0.5) is found to be $\lambda/5.6$, which is less than the light-spot intensity profile-based FWHMs of basis Bessel beams ($\sim \lambda/2.5$).

Fig. R15 The radial variations of the normalized intensity of transverse field (I_{LCP} and I_{RCP}) and the stokes parameter $|S_z|$ ($y = 0$).

(3) For the experimental measurement of the subwavelength feature: The **subwavelength polarization feature is not directly measured experimentally** (due to the lack of setup schemes for directly measure the fine spatially resolved Stokes component S_z fine profile detection); **it is instead derived from the experimentally measured electric fields distribution results under orthogonally polarized states, that is, the spatial distribution S_z was indirectly obtained but affirmatively verified in experiments.**

The detailed experimental procedure is: Firstly, the interference intensity profiles are experimentally captured to recover the complex amplitude of the interference electric fields under orthogonally polarized states; Then, dividing the interference field by the conjugate profile of the coherent plane beam we obtain the complex amplitude of the object beams under x -pol and y -pol (elaborated in revised Supplementary Note 9). The Stokes parameters are derived from the complex amplitudes of electric fields under two orthogonally polarized states (E_x and E_y ; detailed in Supplementary Note 3) to demonstrate this polarization feature with the scale of orientation change being less than the diffraction limit (shown in Fig. 4e; this approach was also applied and verified in previous references, e.g. Ref. [Nature Physics 15, 650-654 (2019)]).

Besides, in this manuscript, we choose the Bessel beams as the modulation basis. In the experiment, we use the object lens with $NA = 0.8$, which is enough to capture and obtain our designed skyrmion based on BBs (Similar to experimental setup in Ref. [Light: Science & Applications 6, e16259-e16259 (2017)], which generates the Bessel beams with spot around $\sim \lambda/3$).

The experimental spatially resolved amplitude features are added and illustrated below:

Fig. R16 The captured intensity profiles in (a) *x*-pol and (b) *y*-pol and the captured interference intensity profiles in (c) *x*-pol and (d) *y*-pol.

(4) The nature of the enhancement: Compared to the spin (Ref. [Nature Physics 15, 650-654 (2019)]) or electric skyrmions (Ref. [Nature Communications 12, 5891 (2021)]), the Stokes skyrmions proposed here can propagate to the far field and its topological features are solely determined by the transverse field, which permits higher degrees of freedom in design and is easy to be captured by CCD (i.e. components experimentally measured by CCD and then derive the Stokes skyrmions). As the subwavelength features of Stokes skyrmion is firstly proposed here and experimentally observed, it can be highly beneficial for super-resolution imaging and sensing in polarization sensitive situation, precision metrology [Ref. Advanced Science, 10, 2205249 (2023)], particle manipulation and etc. It may also lead to other novel and significant physical phenomena that deserve further study.

Comment 4: I suggest that the authors carefully look at the number of self-citations. We all think our own work is important but in such a high impact journal the work should appeal to a wide audience. It does not aid that argument of the reference suggest that only one group are working in the field.

Response 4:

Thank you for the considerate suggestion; we fully agree with your viewpoint. We have removed several less relevant self-citations and included more valuable works from other research groups to better address a broader audience and wider field of study.

The following revisions have been made:

1. Examinations and removal of some previous self citations:

We have carefully reviewed the current citations and removed some articles to ensure that each reference is explicitly necessary.

The following references from the authors have been deleted in the revised manuscript:

Ref. Shen, Y., Yu, B., Wu, H., Li, C., Zhu, Z. & Zayats, A. V. Topological transformation and free-space transport of photonic hopfions. Advanced Photonics 5, 015001 (2023).

Ref. Zdagkas, A., McDonnell, C., Deng, J., Shen, Y., Li, G., Ellenbogen, T. et al. Observation of toroidal pulses of light. Nature Photonics 16, 523-528 (2022).

2. Inclusion of references from other research groups:

We have also added some references from other research groups regarding the application of skyrmions and subwavelength structured light generators (added around line 50, line 196, line 319, as shown below).

Ref. 28 Zhang, X., Liu, G., Hu, Y., Lin, H., Zeng, Z., Zhang, X. et al. Photonic spin-orbit coupling induced by deep-subwavelength structured light. *Physical Review A* **109**, 023522 (2024).

Ref. 29 Zhang, X., Hu, Y., Zhang, X., Li, Z., Chen, Z. & Fu, S. On-Demand Subwavelength-Scale Light Sculpting Using Nanometric Holograms. *Laser & Photonics Reviews* **17**, 2300527 (2023).

Ref. 30 Davis, T. J., Janoschka, D., Dreher, P., Frank, B., Meyer zu Heringdorf, F.-J. & Giessen, H. Ultrafast vector imaging of plasmonic skyrmion dynamics with deep subwavelength resolution. *Science* **368**, eaba6415 (2020).

Ref. 42 Vernon, A. J., Kille, A., Rodríguez-Fortuño, F. J. & Afanasev, A. Non-diffracting polarization features around far-field zeros of electromagnetic radiation. *Optica* **11**, 120-127 (2024).

Ref. 44 Singh, K., Ornelas, P., Dudley, A. & Forbes, A. Synthetic spin dynamics with Bessel-Gaussian optical skyrmions. *Optics Express* **31**, 15289-15300 (2023).